# INTERACTIVE VISUAL EXPLORATION OF LATENT SPACE (IVELS) FOR PEPTIDE AUTO-ENCODER MODEL SELECTION

**Tom Sercu**   **Sebastian Gehrmann**   **Hendrik Strobelt**   **Payel Das**   **Inkit Padhi**

**Cicero Dos Santos**       **Kahini Wadhawan**       **Vijil Chenthamarakshan**

IBM T.J. Watson Research Center, Yorktown Heights, NY 10598
{tom.sercu1,hendrik.strobelt,inkit.padhi,kahini.wadhawan}@ibm.com
gehrmann@seas.harvard.edu
{daspa,cicerons,ecvijil}@us.ibm.com

## ABSTRACT

We present a tool for Interactive Visual Exploration of Latent Space (IVELS) for model selection. Evaluating generative models of discrete sequences from a continuous latent space is a challenging problem, since their optimization involves multiple competing objective terms. We introduce a model-selection pipeline to compare and filter models throughout consecutive stages of more complex and expensive metrics. We present the pipeline in an interactive visual tool to enable the exploration of the metrics, analysis of the learned latent space, and selection of the best model for a given task. We focus specifically on the variational auto-encoder family in a case study of modeling peptide sequences, which are short sequences of amino acids. This task is especially interesting due to the presence of multiple attributes we want to model. We demonstrate how an interactive visual comparison can assist in evaluating how well an unsupervised auto-encoder meaningfully captures the attributes of interest in its latent space.

## 1   INTRODUCTION

Unsupervised representation learning and generation of text from a continuous space is an important topic in natural language processing. This problem has been successfully addressed by variational auto-encoders (VAE) (Kingma & Welling, 2013) and variations, which we will introduce in Section 2. The same methods are also relevant to areas like drug discovery, as the therapeutic small molecules and macromolecules (nucleic acids, peptides, proteins) can be represented as discrete linear sequences, analogous to text strings. Our case study of interest is modeling peptide sequences.

In the VAE formulation, we define the sequence representation as a latent variable modeling problem of inputs $x$ and latent variables $z$, where the joint distribution $p(x, z)$ is factored as $p(z)p_\theta(x|z)$ and the inference of the hidden variable $z$ for a given input $x$ is approximated through an inference network $q_\phi(z|x)$. The auto-encoder training typically aims to minimize two competing objectives: (a) reconstruction of the input and (b) regularization in the latent space. Term (b) acts as a proxy to two real desiderata: (i) "meaningful" representations in latent space, and (ii) the ability to sample new datapoints from $p(x)$ through $p(z)p_\theta(x|z)$. These competing goals and objectives form a fundamental trade-off, and as a consequence, there is no easy way to measure the success of an auto-encoder model. Instead, measuring success requires careful consideration of multiple different metrics. The discussion of the metrics is in Section 2.2, and they will be incorporated in the IVELS tool (Section 5.1 and 5.2).

For generating discrete sequences while controlling user-specific *attributes*, for example peptide sequences with specific functionality, it is crucial to consider *conditional* generation. The most

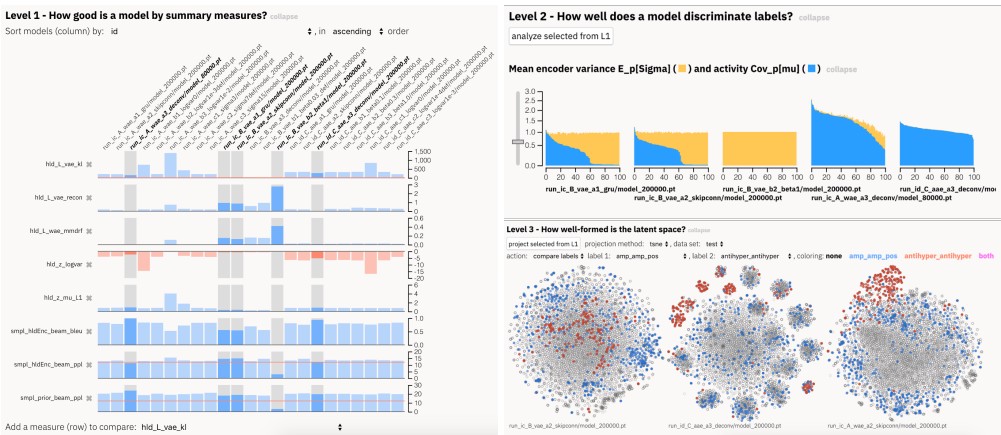

Figure 1: Overview of the IVELS tool. In every stage, we can filter the models to select the ones with satisfactory performance. In the first stage, models can be compared using the static metrics that are typically computed during training (left). In the second stage, we investigate the activity vs noise of the learned latent space (top right) and evaluate whether we can linearly separate attributes (not shown). During the third stage, the tool enables interactive exploration of the attributes in a 2D projection of the latent space (bottom right).

straightforward approach would be limiting the training set to those sequences with the desired attributes. However, this would require large quantities of data labeled with exactly those attributes, which is often not available. Moreover, the usage of those models that are trained on a specific set of labeled data will likely be restricted to that domain. In contrast, unlabeled sequence data is often freely available. Therefore, a reasonable approach for model training is to train a VAE on a large corpus without requiring attribute labels, then leveraging the structure in the latent space for conditional generation based on attributes which are introduced post-hoc. As a prerequisite for this goal, we focus on how $q_\phi(z|x)$ encodes the data with specific attributes. We introduce the encoding of the data subset corresponding to a specific attribute, i.e. the subset marginal posterior, in Section 3. This will be important in the IVELS tool (Section 5.3 and 5.4).

Now that we introduced our models (VAE family), the importance of conditioning on attributes, and our case study of interest (peptide generation), we turn to the focus of our paper. To assist in the model selection process, we present a visual tool for interactive exploration and selection of auto-encoder models. Instead of selecting models by one single unified metric, the tool enables a machine learning practitioner to interactively compare different models, visualize several metrics of interest, and explore the latent space of the encoder. This exploration is building around distributions in the latent space of data subsets, where the subsets are defined by the attributes of interest. We will quantify whether a linear classifier can discriminate attributes in the latent space, and enable visual exploration of the attributes with 2D projections. The setup allows the definition of new ad-hoc attributes and sets to assist users in understanding the learned latent space. The tool is described in Section 5.

In Section 6, we discuss some observations we made using IVELS as it relates to (1) our specific domain of peptide modeling and (2) different variations of VAE models.

## 2 MODELS AND METRICS

### 2.1 MODELS

We approach the unsupervised representation learning problem using auto-encoders (AE) (Hinton & Salakhutdinov, 2006). This class of methods map an input $x$ to continuous latent space $z \in \mathbb{R}^D$ from which the input has to be reconstructed. Regular AEs can learn representations that lead to high reconstruction accuracy when they are trained with sparsity constraints, but they are not suitable for sampling new data points from their latent $z$-space. On the other hand, variational auto-encoders

(VAE) (Kingma & Welling, 2013), which frame an auto-encoder in a probabilistic formalism that constrains the expressivity of $z$, allow for easy sampling. In VAE, each sample defines an encoding distribution $q_\phi(z|x)$ and for each sample, this encoder distribution is constrained to be close to a simple prior distribution $p(z)$. We consider the case of the encoder specifying a diagonal gaussian distribution only, i.e. $q_\phi(z|x) = N(z; \mu(x), \Sigma(x))$ with $\Sigma(x) = \text{diag}[\exp(\log(\sigma_d^2)(x)))]$. The encoder neural network produces the log variances $\log(\sigma_d^2)(x)$. The standard VAE objective is defined as follows (where $D_{\text{KL}}$ is the Kullback-Leibler divergence),

$$\mathcal{L}_{\text{VAE}}(\theta, \phi) = \mathbb{E}_{p(x)} \left\{ \mathbb{E}_{q_\phi(z|x)}[\log p_\theta(x|z)] - D_{\text{KL}}(q_\phi(z|x)||p(z)) \right\}$$

We explore a number of popular model variations we explored for the problem of modeling peptide sequences. With the standard VAE, we observe the same posterior collapse as detailed for natural language in Bowman et al. (2015), meaning $q(z|x) \approx p(z)$ such that no meaningful information is encoded in $z$ space. To address this issue, we introduce a multiplier $\beta$ on the weight of the second KL term (Higgins et al., 2016) i.e. $\beta$-VAE with $\beta < 1$. Alemi et al. (2017) analyze the representation vs. reconstruction trade-offs with a rate-distortion (RD) curve which also provides a justification for tuning $\beta$ to achieve a different trade-off along the RD curve.

We also considered two major variations in the VAE family: Wasserstein auto-encoders (WAE) (Tolstikhin et al., 2017) and adversarial auto-encoders (AAE) (Makhzani et al., 2015). WAE factors an optimal transport plan through the encoder-decoder pair, on the constraint that marginal posterior $q_\phi(z) = \mathbb{E}_{x \sim p(x)} q_\phi(z|x)$ equals a prior distribution, i.e. $q_\phi(z) = p(z)$. This is relaxed to an objective similar to $\mathcal{L}_{\text{VAE}}$ above, but with the per-sample $D_{\text{KL}}$ regularizer replaced by a divergence regularizing a whole minibatch as an approximation of $q_\phi(z)$. In WAE training with maximum mean discrepancy (MMD) or with a discriminator (=AAE), we found a benefit of regularizing the encoder variance as in (Rubenstein et al., 2018; Bahuleyan et al., 2018). For MMD, we used a random features approximation of the Gaussian kernel (Rahimi & Recht, 2007).

In terms of model architectures, the default setting is a bidirectional GRU encoder and a GRU decoder. Skip-connections can be introduced between the latent code $z$ and decoder output (Dieng et al., 2018), which was motivated by avoiding latent variable collapse. Alternatively, one can replace the standard recurrent decoder with a deconvolutional decoder (Zhang et al., 2017), which makes the decoder non-autoregressive, thus forcing it to rely more on $z$.

## 2.2 METRICS

**Metric of interest 1** *The numerical metrics logged during training.*

The starting point of any model evaluation are the typical numerical metrics logged during training. Here we consider the following metrics:

- Reconstruction log likelihood $\log p_\theta(x|z)$ and $D_{\text{KL}}(q(z|x)|p(z))$, computed over the held-out set.
- MMD between $q_\phi(z)$ (average over heldout encodings $q(z|x)$) and the prior $p(z)$.
- Encoder $\log(\sigma_j^2(x))$, averaged over heldout samples and over components $j = 1 \dots D$. Large negative indicates the encoder collapsed to deterministic.
- Reconstruction BLEU score on heldout samples.
- Perplexity evaluated by an external language model for samples from prior $z \sim p(z)$ or from heldout encodings $z \sim q_\phi(z|x)$.

All of our sample-based metrics (BLEU, perplexity) are using beam search as the decoding scheme.

**Metric of interest 2** *Activity and encoder variance.*

We use *(unit) activity* as a proxy to investigate how many latent dimensions are encoding useful information about the input. We will extend the concept to evaluate whether the marginal posterior $q_\phi(z)$ is far from the prior.

*Active units* are defined as the number of dimensions $d = 1 \dots D$ where the *activity* $A_d = \text{Cov}_{p(x)}[\mathbb{E}_{q_\phi(z|x)}[z_d]]$ is above a threshold (Burda et al., 2015). The activity tells us whether the

encoder mean $\mu_\phi(x)$ varies over observations. To expand on this notion, we follow Kumar et al. (2017) and focus on the marginal posterior $q_\phi(z)$. In the usual parametrization where the encoder is used to specify the mean $\mu_\phi(x)$ and diagonal covariance $\Sigma_\phi(x)$ of a Gaussian distribution, the total covariance is given by:

$$\text{Cov}_{q_\phi(z)}[z] = \text{Cov}_{p(x)}[\mu_\phi(x)] + \mathbb{E}_{p(x)}[\Sigma_\phi(x)]$$

This tells us the $d$-th diagonal element of the covariance matrix $\text{Cov}_{q_\phi(z)}[z]_{d,d}$ is the sum of the activity $A_d$ and the average encoder variance $\sigma_d^2(x)$ (i.e. how much noise is injected along this dimension on average). To satisfy $q_\phi(z) = p(z)$ we need at least the first and second order moments of $q_\phi(z)$ to match those of $p(z)$, and thus we need $\text{Cov}_{q(z)}[z] \approx I$. Inspecting the two terms of $\text{Cov}_{q_\phi(z)}[z]$ along the diagonal can thus tell us (i) an obvious way to violate closeness to the prior and (ii) whether the covariance is dominated by activity or encoder uncertainty.

## 3 ANALYZING ATTRIBUTES IN LATENT SPACE

In this work, we focus on unconditionally trained VAE models. Even though several approaches have been proposed to incorporate attribute or label information during VAE training (Kingma et al., 2014; Sohn et al., 2015), they require all labels to be present at training time, or require a strategy to deal with missing labels to enable semi-supervised learning. To avoid this, we follow Engel et al. (2017) and aim to train a sequence auto-encoder model unconditionally and rely on the structure of the latent $z$-space to define the attributes post-hoc. This process also eliminates the need for retraining the VAE when new labels are acquired or new attributes are defined. Specifically, from the interactivity standpoint, this enables end users or model trainers to interactively specify new attribute groups or subsets of the data. We aim to enable attribute-conditioned generation by understanding the learned latent space of a model.

Let the attributes be $\mathbf{a} = [a_1, \ldots a_n]$, with each attribute $a_i$ taking a value $y \in A_i$ (typically: positive, negative, not available). Since the probability of the intersection of $n$ attributes can be small or zero, we focus on conditioning on a single attribute at a time. In general, we define a subset $S$ of our dataset as those datapoints where attribute $a_i = y$ and denote the corresponding distribution as $p^S(x)$ or $p^{a_i=y}(x) = p(x|a_i = y)$. By focusing on the subset $S$ defined by selecting on $a_i = y$, we have the flexibility to define new subsets online with the same notation.

In the auto-encoder formulation and variants we discussed, an important object is the maginal posterior $q_\phi(z) = \mathbb{E}_{x \sim p(x)} q_\phi(z|x)$, which is introduced as the aggregate posterior by Makhzani et al. (2015). This distribution is central to the analysis of Hoffman & Johnson (2016) as well as WAE which relies on $q_\phi(z) = p(z)$.

Let us now define the marginal posterior for subset $S$ with distribution $p^S(x)$:

$$q_\phi^S(z) = \mathbb{E}_{x \sim p^S(x)} q_\phi(z|x)$$

The *subset marginal posterior* is an essential distribution for our analysis as it tells us how the distribution corresponding to a specific attribute is encoded in the latent space. Since we aim to be able to sample from $q_\phi^S(z)$ conditionally, we require the distribution to have two properties. First, $q_\phi^S(z)$ needs to be distinct from the background distribution $q_\phi(z)$ (Aim 1). We found that the underlying data-generating distributions of labeled and unlabeled data do not necessarily match for biological applications. Since there might be an underlying reason why a data point has not been labeled, a model should learn which points are of interest for a particular attribute. The second aim is that $q_\phi^S(z)$ should have sufficient regularity to capture the distribution with simple density modeling (Aim 2). Being able to discriminate between different attribute labels within the same category is crucial when aiming to generate sequences with a particular property. To be able to analyze arbitrary subsets in an interactive exploration of $q_\phi^S(z)$, we focus on the following two metrics of interest.

**Metric of interest 3** *Performance of a simple linear classifier (logistic regression) trained to discriminate between $q_\phi^S(z)$ vs $q_\phi^{S'}(z)$, with $S \neq S'$.*

This metric addresses the question of whether we can easily discriminate the subset $S$ (corresponding to attribute $a_i$) in the latent space.

To address the aims introduced above, we consider two approaches to define $S$ and $S'$:

- $a_i$ available vs $a_i$ not available (Aim 1).
- $a_i = y$ vs $a_i = y'$ two different attribute labels (e.g., $y =$ positive, $y' =$ negative) (Aim 2).

**Metric of interest 4** *2D projections of the full marginal posterior $q_\phi(z)$ and the place of each subset marginal posterior $q_\phi^S(z)$ in it.*

While static metrics can provide an intuition for the quality of the latent space, we further aim to analyze the well-formedness of the space. Thus, we investigate how attributes cluster visually in 2D projections for different models.

## 4 CASE STUDY: PEPTIDE MODELING

Peptides are single linear chains of amino acids. We consider sequences that are composed of twenty natural amino acids, i.e., our vocabulary consists of 20 different characters. The length of sequences is restricted to $\leq 25$. Depending on the amino acid composition and distribution, peptides can have a range of biological functions, e.g., antimicrobial, anticancer, hormone, and are therefore useful in therapeutic and material applications.

Latent variable models such as VAEs have been successfully applied for capturing higher-order, context-dependent constraints in biological peptide sequences (Riesselman et al., 2017), for semi-supervised generation of antimicrobial peptide sequences (Das et al., 2018), and for revealing distinct cancer-specific methylation patterns of DNA sequences (Titus et al., 2018). Gómez-Bombarelli et al. (2018) have used VAE models for automatic design of chemicals from SMILES strings.

In this study, we focus on comparing the latent spaces learned by modeling the peptide sequences at the character level by the use of VAE and its variants. Furthermore, we investigate the feasibility of using the latent space learned using a large corpus of unlabeled sequences to track representative distinct functional families of peptides. For this purpose, we mainly focus on five biological functions or attributes of peptides, *i.e.* antimicrobial (AMP), anticancer, hormonal, toxic and antihypertensive.

Frequently, these peptides are naturally expressed in a variety of host organisms and therefore are identified as the most promising alternative to conventional small molecule drugs. As an example, given the global emergence of multidrug-resistant bacterias or "superbugs" and a dry discovery pipeline of new antibiotics, AMPs are considered as exciting candidates for future infection treatment strategies. Water solubility is also considered as an additional attribute.

Our labeled dataset comprises sequences with different attributes curated from a number of publicly available databases (Singh et al., 2015; Gogoladze et al., 2014; Khurana et al., 2018; Bhadra et al., 2018; Gupta et al., 2013). Below we provide details of the labeled dataset:

- Antimicrobial (8683 positive, 6536 negative);
- Toxic (3149 positive, 16280 negative);
- Hormone (569 positive);
- Antihypertensive (1659 positive);
- Anticancer (504 positive);
- Water-soluble (195 positive, 31 negative).

The labeled data are augmented with the unlabeled dataset from Uniprot-SwissProt and Uniprot-Trembl (Consortium, 2007), totaling $\sim 1.7$ M points. The data is split into separate train, valid, and test sets corresponding to 80%, 10% and 10% of the data, respectively. Data for which an attribute is present is upsampled with a factor $20\times$ during training.

## 5 INTERACTIVE VISUAL EXPLORATION OF LATENT SPACE (IVELS) FOR MODEL SELECTION

Our tool aims to support the role of a model trainer as described by Strobelt et al. (2018). This role does not assume extensive domain knowledge, but an understanding of the model itself. As such, we

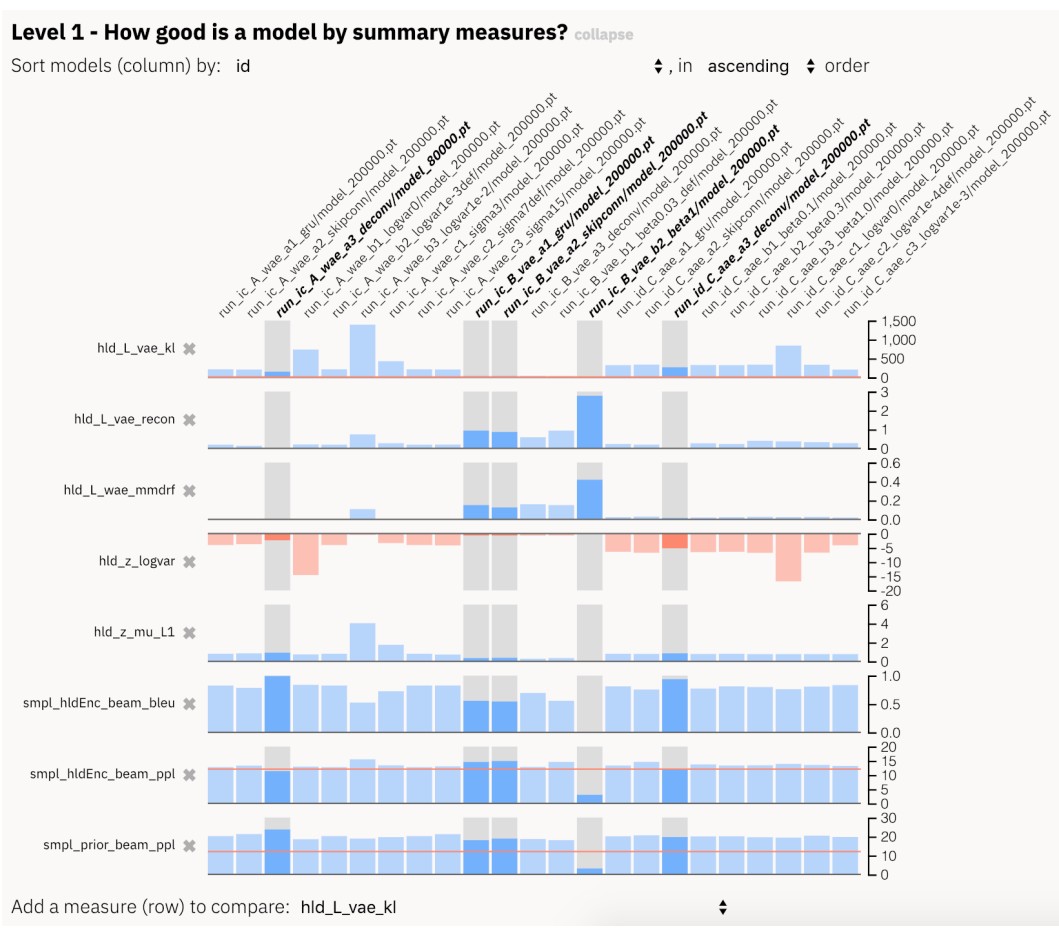

Figure 2: Level 1. Numerical metrics logged during training. The tool allows users to add arbitrary metrics that are logged for a model. The red line in perplexity plots represents baseline value.

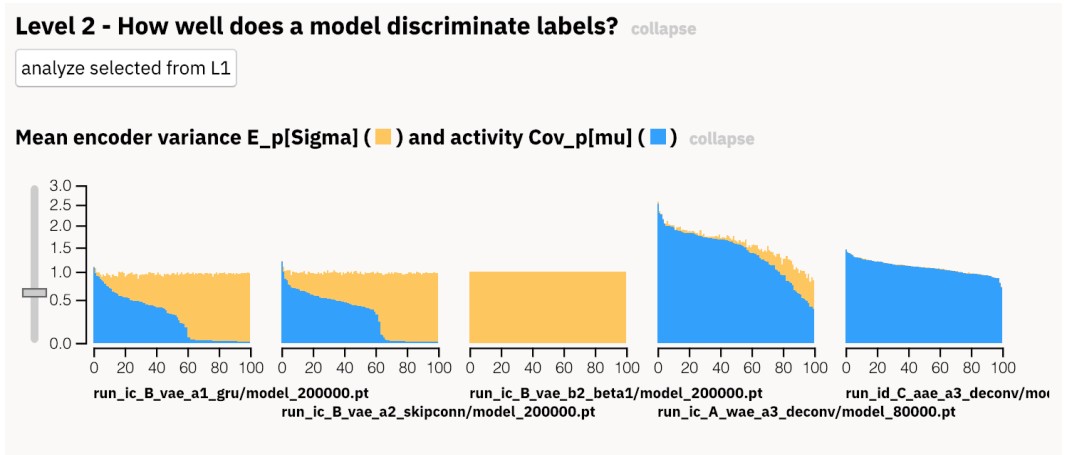

Figure 3: Level 2.1. Activity (blue) and mean encoder variance (orange). These are the diagonals of respectively $\mathrm{Cov}_{p(x)}[\mu_\phi(x)]$ and $\mathbb{E}_{p(x)}[\Sigma_\phi(x)]$.

limit the visual elements of the tool to those that do not evaluate sequences at a granular peptide-level. As a consequence, the tool aims to visualize high-level information that is captured during training and iteratively focuses down to the medium-level, where we evaluate attributes.

Specifically, we introduce a three-level pipeline that enables a user to conduct a cohort-analysis of all the models they have trained. During each level, the user can filter the remaining models based on information provided by the tool. The iterative filtering process further allows successively more computationally expensive analyses. The subsections here follow the measures highlighted as "Metric of interest" in the previous sections. The details of the models appearing in the figures are found in Appendix A.

### 5.1 Numerical metrics

The first level (Figure 2) is a side by side view of a user-specified subset of the metrics logged during training (rows) across multiple models (columns). In our example, we select only the model checkpoints from the final epoch of each training run. These metrics would be typically inspected by a model trainer in graphs over time, for example in tensorboard. However, while graphs over time excel at communicating results for a single metric, comparing models across multiple metrics is challenging and time-consuming. By showing an arbitrary number of metrics at once, the IVELS tool enables the selection of promising models for further analysis.

The tool further simplifies the selection process through the ability to sort the columns by different metrics. Sorting makes it easy to select models that achieve at least a certain performance threshold.

### 5.2 Active Units and Covariance diagonal

Figure 3 presents the encoding of Metric of Interest 2, i.e. the diagonal of $\text{Cov}_{q_\phi(z)}[z]$. For interpretability, we sort the latent dimensions according to decreasing activity. This visual representation allows inspection of the balance between activity (encoder mean changing depending on observations), and average encoder variance (how much the noise is dominating).

We discuss the observations for different models in Section 6.2.

### 5.3 Discriminating attributes

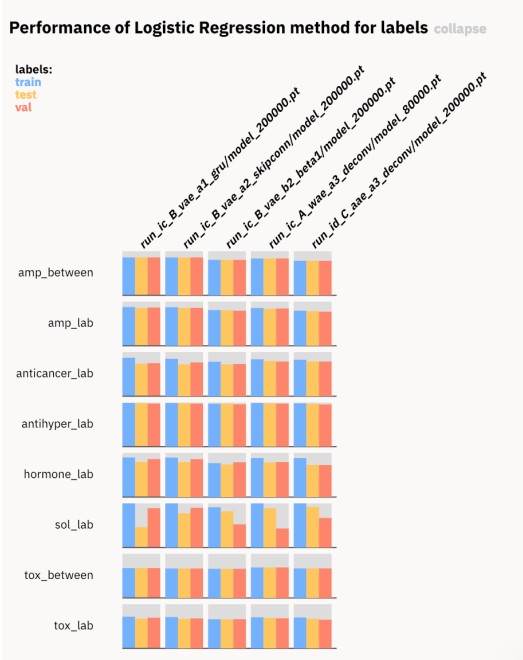

Given that we established that $z$ is actively encoding information in the first part of level 2, the second part of level 2 aims to evaluate whether we can linearly separate attributes within the learned space. This is a prerequisite for level 3, which assumes that the encodings can be related to the attributes in a meaningful way.

Figure 4 presents the results for the models across the attributes that are available in the dataset. Following the Metric of Interest 3, we differentiate between positive and negative labels (indicated by `lab`) and labeled and unlabeled samples (indicated by `between`). For each of these scenarios, we train a logistic regression on $z$ of the training set and evaluate it on the training, validation and test sets. To account for a dynamic number of different labels, the results for `lab` are the accuracy, whereas `between` is measured in AUC. The $y$ axis is scaled in $[0, 1]$.

Figure 4: Level 2.2. Attribute discriminability in the latent space. Despite being trained fully unsupervised, all models successfully encode multiple attributes in $z$.

## 5.4 Projections of the latent space

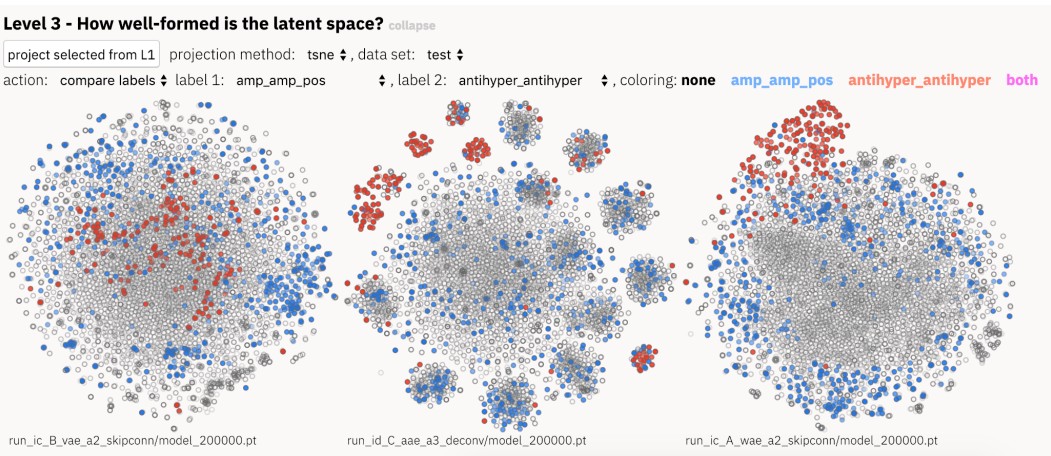

Figure 5: Level 3. 2D projections of the latent space, showing samples from the marginal posterior $q_\phi(z)$. Subsets of interest ($q_\phi^S(z)$) are color-coded.

We allow the user to select either t-SNE (Maaten & Hinton, 2008) or linear projection on axes of interest (PCA). To visualize the different attributes, we enable color-coding in two modes: (1) show labels and (2) compare labels. "Show labels" will color-code the different values a single attribute can assume, in our case positive and negative. "Compare labels" allows to select two different attributes with a specific value. Using this mode, we can for example examine whether there is a section of the latent space that has learned both soluble and AMP peptides. Should a data point have been annotated with both of the selected values, it is color-coded with a separate color.

## 6 Discussion and Results

### 6.1 Peptide Case Study

From stage 2 (Fig. 4) it is encouraging to see that in this unconditionally trained model the different attributes are salient as judged by a simple linear classifier trained on the latent dimensions. Some general trends appear. We can observe generally high performance across all models, which is promising for conditional sampling. One difference of note is that the $\beta$-VAE and AAE models perform worse on the AMP attribute. The discriminators for attributes with limited annotation, specifically water-solubility, overfit on the training set which indicates a need for further annotation efforts. The results further demonstrate that toxicity is more challenging than the remaining attributes despite being the second-most common attribute in the dataset. These results set the stage for further investigation of the latent space. Fig. 5 shows the tSNE projections of the latent space learned using three different models (Level 3). Two distinct attributes, positive antimicrobial (amp_amp_pos, in blue) and antihypertensive (antihyper_antihyper, in red), are also shown. Interestingly, these two attributes are more well-separated in the latent space obtained using AAE and WAE (Fig. 5, middle and right) compared to that from VAE (Fig. 5, left). From the biological perspective, antihypertensive peptides are distinct in terms of length, amino acid composition, and mode of action. Antimicrobial peptides are typically longer than the antihypertensive ones. Also, antimicrobial peptides function by disrupting the cell membrane, while antihypertension properties originate from enzyme inhibition (Majumder & Wu, 2015).

### 6.2 VAE variations

From the $\text{Cov}_{q_\phi(z)}[z]$ plot (Level 2.1, Figure 3), we can observe that the VAE (column 3) suffers from posterior collapse, since it has no active units. We can further see that the $\beta$-VAE (column 1 and 2) address the collapse issue and about half of its dimensions in $z$ space are active. Interestingly, the dimensions that are not active become dominated by encoder variance, such that the total variance for

each dimension is close to 1. The skip-connection added to the GRU (2nd column) lead to a slightly higher activity around the tail of the active dimensions, though the difference is minor. The WAE and AAE (column 4 and 5) have relatively little encoder variance, meaning they are almost deterministic. Notably, the WAE covariance is furthest away from the prior.

From the t-SNE plots (Fig. 5) we see the WAE and AAE show good clustering of the attributes like positive antimicrobial and antihypertensive, showing that the latent space is clearly capturing those attributes, even though they were not incorporated at training time.

## 7    CONCLUSION

We presented a tool for Interactive Visual Exploration of Latent Space (IVELS) for model selection focused on auto-encoder models for peptide sequences. Even though we present the tool with this use case, the principle is generally useful for models which do not have a single metric to compare and evaluate. With some adaptation to the model and metrics, this tool could be extended to evaluate other latent variable models, either for sequences or images, speech synthesis models, etc. In all those scenarios, having a usable, visual and interactive tool for model architects and model trainers will enable efficient exploration and selection of different model variations. The results from this evaluation can further guide the generation of samples with the desired attribute(s).

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

# Appendix

## A  MODEL CONFIGURATIONS

| AE Type | Prefix | Variation | Description |
|---|---|---|---|
| wae | A_wae | a1_gru | GRU Enc-Dec |
| | | a2_skipconn | GRU Enc-Dec with skip connection |
| | | a3_deconv | GRU Enc & Deconvolutional Dec |
| | | b**[1,2,3]**_logvar**[0,1e-3, 1e-2]** | $z$-space logvar regularization |
| | | c**[1,2,3]**_sigma**[3,7,15]** | RBF kernel bandwidth |
| vae | B_vae | a1_gru | GRU Encoder and Decoder |
| | | a2_skipconn | GRU Enc-Dec with skip connection |
| | | a3_deconv | GRU Enc & Deconvolutional Dec |
| | | b**[1,2]**_beta**[0.03, 1]** | KL weight annealing |
| aae | C_aae | a1_gru | GRU Encoder and Decoder |
| | | a2_skipconn | GRU Enc-Dec with skip connection |
| | | a3_deconv | GRU Enc & Deconvolutional Dec |
| | | b**[1,2,3]**_beta**[0.1,0.3,1.0]** | Adversarial regularization loss weight |
| | | c**[1,2,3]**_logvar**[0, 1e-4, 1e-3]** | $z$-space logvar regularization |

Table 1: Variations in Model Cofigurations

We adopted in our experiments three different types of autoencoding approaches: $\beta$-VAE, Wasserstein auto-encoder (WAE) and Adversarial auto-encoder (AAE). For each of these AEs the default is bi-GRU encoder and GRU decoder. Table 1 summarizes these AEs (prefix: *A_wae, B_vae, C_aae*) along with their variations (prefix: *a1-a3, b1-b3, c1-c3*).

For the encoder, we used a bi-directional GRU with hidden state size of 80. For the decoder we tried three major variations: a) plain GRU decoder (prefix:*a1_gru*) b) GRU decoder with skip-connections between $z$ and the output (prefix:*a2_skipconn*) c) deconvolutional decoder (prefix:*a3_deconv*). Our deconv decoder contains 3 strided deconvolutional layers, and three convolutional layer arranged as follows: *deconv1→deconv2→conv1→conv2→deconv3→conv3*. Across this architecture we used 100 as the number of convolutional filters and a convolving kernel size of 4. In all these frameworks, the latent capacity was set at $D = 100$ and we fixed the maximum character-level sequence length to 25.

For VAE we used KL term annealing $\beta$ from 0 to 0.03 by default. We also present an unmodified VAE with KL term annealing $\beta$ from 0 to 1.0 (prefix:*b2_beta1*). For WAE, we found that the random features approximation of the gaussian kernel with kernel bandwidth $\sigma = 7$ (prefix:*c2_sigma*) to be peforming the best. For comparison sake, we have included variations with $\sigma$ values of 3 and 15 too (prefix:*c1_sigma,c3_sigma*). As described in 2.1, the inclusion of $z$-space noise logvar regularization avoided collapse to a deterministic encoder. The prefixes *b1_logvar, b2_logvar, b3_logvar* translate into different regularization weights used, of which $1e-3$ have the most desirable behavior on the metrics. The default value of $\sigma = 7$ and logvar regularizer of 1e-3 is used across the variations consisting of gru encoder-decoder, skip-connection and deconv. decoder (prefix:*a1-a3*). For AAE, we tried annealing weight ($beta$) on the adversarial regularization loss from 0.0 to 0.1/0.3/1.0 (prefixes:*b1_beta0.0, b2_beta0.3 and b2_beta1.0*) and also, applied the logvar regularization on the $z$. The default value of 0.3 for $\beta$ and 1e-4 for logvar regularization was used for experiments with different architectural variations.

