# OpenReview forum: "Interactive Visual Exploration of Latent Space (IVELS) for peptide auto-encoder model selection"
_ICLR.cc/2019/Workshop/DeepGenStruct — DeepGenStruct 2019_

### Official Review · AnonReviewer2 · 2019-04-08
**Interesting visualization tool for generative models**

**Rating:** 3
**Confidence:** 2

**Review:**

This paper proposed a tool for visualizing the latent spaces for generative models. The authors demonstrated this tool by applying this tool to show some visualizations of some auto-encoders (VAE, WAE and AAE).

Pros:

1. The paper is easy to follow and the visualization results are clear. It is easy to understand the metrics from the figures of the visualization tool.

2. The visualized metrics are useful and can show the differences between the WAE and the AAE compared with the VAE, which means that the visualizations are successful.

3. The visualization tool can also analyze the attributes in the latent space, which is useful.

Cons:

1. It is better if more generative models (e.g. GAN) can be studied using this tool.

---

### Official Review · AnonReviewer1 · 2019-04-14
**tool for analysing vae over discrete data**

**Rating:** 3
**Confidence:** 2

**Review:**

The paper presents a tool to analyse various aspects of model trained using the VAE (amortised VI) framework for discrete data. VAE framework is known to be prone to several learning related issues such as slow convergence, posterior collapse, etc. therefore such a tool could provide a significant insight in tuning the model as well as selecting a model that best suits the needs.

---

### Decision · Program_Chairs · 2019-04-19
**Acceptance Decision**

**Decision:**

Accept

**Comment:**

Accepted